# Proliferation, Migration and Invasion of Breast Cancer Cell Lines Are Inhibited by 1,5-Disubstituted Tetrazol-1,2,3-triazole Hybrids through Interaction with p53

**DOI:** 10.3390/molecules28227600

**Published:** 2023-11-15

**Authors:** Marisol Moreno-Perea, Abel Suárez-Castro, Ixamail Fraire-Soto, Jessica Lizbeth Sifuentes-Padilla, Rosalinda Gutiérrez-Hernández, Claudia Araceli Reyes-Estrada, Yamilé López-Hernández, Carlos J. Cortés-García, Luis Chacón-García, Angelica Judith Granados-López, Jesús Adrián López

**Affiliations:** 1Laboratorio de microRNAs y Cáncer, Universidad Autónoma de Zacatecas, Av. Preparatoria S/N, Agronómica, Campus II, Zacatecas 98066, Zacatecas, Mexicoixamail13@gmail.com (I.F.-S.); jsifuentespadilla@gmail.com (J.L.S.-P.); 2Laboratorio de Diseño Molecular, Instituto de Investigaciones Químico-Biológicas, Universidad Michoacana de San Nicolás de Hidalgo, Ciudad Universitaria, Morelia 58033, Michoacán, Mexico; abel.suarez@umich.mx (A.S.-C.); ccortes@umich.mx (C.J.C.-G.); 3Unidad Académica de Enfermería, Universidad Autónoma de Zacatecas, Campus Siglo XXI, Edificio L-1, Segundo Piso, Carretera Zacatecas-Guadalajara Km 6, Ejido La Escondida, Zacatecas 98160, Zacatecas, Mexico; rosalinda@uaz.edu.mx; 4Maestría en Ciencias de la Salud con Especialidad en Salud Pública, Unidad Academica de Medicina Human, UAZ, Campus Siglo XXI, Edificio L-1, Segundo Piso, Carretera Zacatecas-Guadalajara Km 6, Ejido La Escondida, Zacatecas 98160, Zacatecas, Mexico; 5Laboratorio de Metabolómica y Proteómica, Cátedra CONACYT, Unidad Académica de Ciencias Biológicas, Universidad Autónoma de Zacatecas, Av. Preparatoria S/N, Agronómica, Campus II, Zacatecas 98066, Zacatecas, Mexico

**Keywords:** 1,5-disubstituted tetrazole, 1,4-disubstituted-1,2,3-triazoles, hybrid compounds, cell proliferation, cell migration, cell invasion

## Abstract

The anticarcinogenic potential of a series of 1,5-disubstituted tetrazole-1,2,3-triazole hybrids (T-THs) was evaluated in the breast cancer (BC)-derived cell lines MCF-7 (ER+, PR+, and HER2−), CAMA-1 (ER+, PR+/−, and HER2−), SKBR-3 (ER+, PR+, and HER2+), and HCC1954 (ER+, PR+, and HER2+). The T-THs **7f**, **7l**, and **7g** inhibited the proliferation of MCF-7 and CAMA-1, HCC1954, and SKBR-3 cells, respectively. The compounds with stronger effect in terms of migration and invasion inhibition were **7o**, **7b**, **7n**, and **7k** for the CAMA-1, MCF-7, HCC1954, and SKBR-3 cells respectively. Interestingly, these T-THs were the compounds with a fluorine present in their structures. To discover a possible target protein, a molecular docking analysis was performed for p53, p38, p58, and JNK1. The T-THs presented a higher affinity for p53, followed by JNK1, p58, and lastly p38. The best-predicted affinity for p53 showed interactions between the T-THs and both the DNA fragment and the protein. These results provide an opportunity for these compounds to be studied as potential drug candidates for breast cancer treatment.

## 1. Introduction

The aromatic nitrogenated heterocycles, mainly 1,5-disubstituted tetrazoles (1,5-DS-T) and 1,4-disubstituted-1,2,3-triazoles, have become increasingly popular in early drug discovery projects, and this popularity is mainly due to their presence in many bioactive compounds and FDA drug approval [1,2,3]. However, developing efficient synthetic strategies for obtaining these heterocycles has proven challenging for organic chemists [4,5]. In recent years, the molecular hybridization of these heterocycles using powerful synthetic techniques such as isocyanide-multicomponent reactions ha been explored to overcome this challenge. Cortés-García’s research group has reported success in this field [6,7,8]. Thus, recently, we reported a new high-order multicomponent reaction that can be used to synthesize a series of 1,5-disubstituted tetrazole-1,2,3-triazole hybrids involving a one-pot, six-component reaction and investigated these hybrids’ cell proliferation effect on MCF-7 cells [8], a commonly used model of breast cancer (BC) (Figure 1). 

With 684,996 deaths registered annually [9], BC is one of the most frequent carcinomas and deathliest pathologies worldwide [10,11]. The impact of this pathology is growing around the world, and its study, regarding new compounds with possible antitumoral effects, is imperative. Therefore, continuing with the characterization of hybrid compounds, we used four cell lines to simulate the luminal A and B stages of breast cancer (BC), which are based on PR, ER, and HER2 markers expression [12]. It has been established the positive expression of PR and ER and negative expression of HER2 for luminal A (MCF-7 (ER+, PR+, and HER2−) andCAMA-1 (ER+, PR+/−, and HER2−)), while positive expression of PR and ER and positive expression of HER2 has been observed for luminal B (SKBR-3 (ER+, PR+, and HER2+) and HCC1954 (ER+, PR+, and HER2+)). Based on this in vitro model, we analyzed proliferation, migration, and invasion inhibition in response to hybrid compounds. 

## 2. Results and Discussion

Thirteen hybrid compounds with structural diversity and different chemical composition and yields were synthesized previously (Figure 1). The identities of all the compounds were confirmed using ^1^H and ^13^C NMR spectroscopy as well as HRMS in our previous work (Aguilar-Morales et al. [8]; Appendix A). BC kills more women around the world than any other cancer, making this pathology one of most important carcinomas in our time. One of the causes of cancer remission is chemoresistance generated by the overexpression of genes involved in transport and DNA repair among other processes [13,14]. Bigger tumors cause more damage than smaller ones and are related to metastasis leading to death among all patients. Consequently, compounds that reduce tumor size and inhibit the metastasis process are of great interest for basic and translational research. Therefore, compounds **7a–o** were tested to evaluate their capacity to inhibit cell proliferation, migration, and invasion in an in vitro BC model of ER, PR, and HER2-positive or -negative markers. The cell lines MCF-7 (ER+, PR+, and HER2−), CAMA-1 (ER+, PR+/−, and HER2−), (SKBR-3 (ER+, PR+, and HER2+), and HCC1954 (ER+, PR+, and HER2+) are models of invasive ductal breast carcinoma representing 70 to 80% of all invasive BCs and can be divided into luminal A (MCF-7 and CAMA-1) and luminal B (SKBR-3 and HCC1954) BC subtypes.

Previously, we reported that the inhibition of the cell proliferation of MCF-7 cells was primarily achieved by the following compounds in decreasing order of effect: **7f**, **7l**, **7n**, **7b**, **7d**, **7h**, **7j**, **7k**, and **7e**; in contrast, **7a**, **7c**, **7i**, **7g,** and **7o** did not show a significant effect [8]. Herein, we observed that CAMA-1 cells were inhibited by the following compounds: **7f**, **7l**, **7b**, **7j**, **7a**, **7h**, **7k**, **7g**, **7n**, **7m**, and **7o;** furthermore, **7c**, **7d**, **7e**, and **7i**, did not elicit an effect (Figure 1a). Interestingly, it should be noted that some compounds’ effects were shared in the MCF-7 and CAMA-1 cells, probably due to the similarity of their molecular markers. On the other hand, HCC1954 cells were inhibited by all compounds, except **7o**, in the following order: **7g**, **7b**, **7h**, **7e**, **7f**, **7a**, **7j**, **7n**, **7l**, **7d**, **7k**, **7i**, **7c**, and **7m** (Figure 1b). The compounds **7l**, **7f**, **7n**, **7h**, **7k**, **7j**, **7a**, **7g**, and **7b** inhibited the cell proliferation of SKBR-3, while **7e**, **7i**, **7o**, **7m**, **7c**, and **7d** did not elicit a significant effect (Figure 1c). The compounds that inhibited HCC1954 and SKBR-3 were dissimilar in order and strength. Therefore, an association between the molecular markers and the observed effect is not supported by our results. 

Noticeable effects of **7f** on the MCF-7 and CAMA-1 cells, **7g** on the HCC1954 cells, and **7l** on the SKBR-3 cells were observed (Figure 1 and Table 1). Interestingly, **7f** inhibited HCC1954 and SKBR-3 cell proliferation to the second-highest degree, indicating the wider scope of application of this compound. Interestingly, compound **7l** inhibited all the BC cell lines tested, while **7g** was specific for HCC1954 cells, which is a remarkable characteristic of a compound intended for application to a certain BC genotype (Table 1). New alternative compounds could be useful for Taxol- and/or Etoposide-resistant tumor therapy, either alone or in combination with baseline anticancer drugs [15]. Zhang et al. reported an inhibition of 70% in MCF-7 cells with 100 µM of etoposide [16], while Abzianidze reported an IC_50_ of 9.6 [17]. The T-THs with greater activity tested in this work oscillated between 10.3 and 34 µM, showing the antitumorigenic potential of these compounds that could be used in combination or as substitutes for the drugs commonly used. The proliferation-inhibitory effect of our compounds was found to be similar to that of other groups of compounds. The activity of the tested T-TH compounds was shown to be similar to that of other groups of compounds, such as the 18 novel tetrazole-based diselenides and seleoquinones reported by Shaaban and co-workers, which were synthesized via an Ugi-azide reaction and assayed on MCF-7 cell lines and whose proliferation inhibition was ranked from 14 to 78 µM [18]. Our results are also comparable to those found by Nagarapu’s research group, which synthesized a series of 1,2,3-triazole-tethered chalcone acetamides that were ranked from an IC_50_ of 9.76 to 98 µM [19], or to carboxyamido-triazoles that exhibited an IC_50_ of 16 µM on MCF-7 cells [20]. In our work, we reported an IC_50_ of 19.76 µM toward MCF-7 caused by **7f**, while an IC_50_ of 33.34 was attained in CAMA1 cells. More favorably, the T-THs **7l** and **7g** induced an IC_50_ of 18.63 and 10.28 in SKBR-3 and HCC1954 cells, respectively (Table 1). Interestingly, the BC cells’ proliferation and/or survival responses to several compounds have been tested. 

It has been suggested that the effect on cell proliferation is dependent on cell type and the particular expression of genes and metabolites. Tetrazoles have been reported to be able to inhibit cell proliferation by targeting PRMT, COX2, and tubulin polymerization in SAR prediction [21]; however, experimental demonstrations are still missing. On the other hand, triazole derivatives have been proven to affect mitochondrial metabolism, leading to apoptosis [22]. Compounds with triazole and tetrazole combinations in their structures can also hinder additional cancer hallmarks. Therefore, to further characterize the antitumoral effect of the compounds on the BC cell lines, we explored their effect on the cell processes of migration and invasion. 

Interestingly, for the MCF-7 and CAMA-1 cells, the compound that inhibited a greater share of cell proliferation did not exert an effect on migration or invasion (Figure 1 and Figure 2). Instead, compound **7b** induced migration impairment on the MCF-7 cells (Figure 2a,b), while compound **7m** exerted the strongest effect on migration inhibition of the CAMA-1 cells. In a very specific manner, compounds **7g** and **7l**, which induced a proliferation impediment, did not elicit an important effect on the HCC1954 and SKBR-3 cell lines, respectively, in terms of cell migration. Interestingly, cell migration and invasion were affected by compounds **7a** and **7n** in the HCC1954 cells, while compounds **7b** and **7k** produced the stoutest effect on SKBR-3 cells. Intriguingly, the T-TH compounds induced the inhibition of specific cell processes in particular cell lines, as was the case for compound **7g** that inhibited HCC1954 cell proliferation exclusively but suppressed migration and invasion on MCF-7 and CAMA-1 cells. The compounds that affected the proliferation as well as migration and invasion of the four cell lines tested were **7b**, **7n**, and **7k,** suggesting that they are the most complete ones attacking cancer cells, independently of their genomic background. The effects shown in our results are similar to those found in other works like that of Lambert et al., where they tested carboxyamido-triazole, showing cell proliferation and invasion inhibition [20]. Several targets have been found to impede migration and invasion. Zheng et al. found that 1,2,3-triazole-dithiocarbamate hybrids interfered with migration and invasion by inhibiting Lysine-specific demethylase 1 (LSD1), a histone demethylase [23]. Regarding the targets of tetrazol compounds, research incorporating 6-tetrazolyl-substituted sulfocoumarins showed the inhibition of carbonic anhydrase (CA) isoforms, the cytosolic hCAs I and II and the transmembrane, and tumor-associated hCA IX and XII. The tetrazole-substituted sulfocoumarins showed effective inhibition against the two transmembrane CAs [24]. Additionally, the seven-amino-[1,2,4]triazolo[4,3-f]pteridinone and 7-aminotetrazolo[1,5-f]pteridinone derivatives were found to target Polo-Like Kinase 1 (PLK1) based on docking and enzymatic assays [25]. Therefore, it should be expected that the compounds could target one or more proteins or metabolites to induce their effects. In this context, modifications to the structure of compounds **7b**, **7f**, **7g**, **7k**, **7l**, **7m**, and **7n** are underway to achieve better results in terms of proliferation, migration, and invasion inhibition as well as the characterization of possible targets based on molecular docking in order to understand the mechanisms of the effects observed on the hallmarks of cancer in these cell lines.

To elucidate the possible proteins involved in the biological activity of the 1,5-disubstituted tetrazole-1,2,3-triazoles, molecular docking studies were carried out with p53, p38, p58, and JNK1 proteins involved in carcinogenesis. These in silico studies were performed in a non-targeted manner at a specific site of each molecule, allowing the ligands to freely interact with each one of the proteins in their rigid three-dimensional conformations. The predicted free energy results, as well as the affinity constants of each compound, are shown in Table 2.

In general, the 1,5-disubstituted tetrazole-1,2,3-triazoles presented a higher affinity for p53, followed by JNK1, p58, and lastly p38. For the case of p53 as a receptor, it can be observed that the compounds with the best-predicted affinity were **7c**, **7d**, **7g**, and **7f**, with interaction free energy values of −11.26 kcal/mol; −11.26 kcal/mol; −10.71 kcal/mol; and −10.66 kcal/mol, respectively. Regarding the interactions and the binding modes of the 1,5-disubstituted tetrazole-1,2,3-triazoles, an analysis of the main non-covalent interactions between all the compounds and the four receptors was carried out, highlighting compound **7f** (Figure 3).

The amino acid residues His179, Cys176, and Cys242 are necessary for the stabilization of p53 through zinc ions. Compound **7f** could play an important role through interactions with p53 and zinc ions contributing to its stability via its hydrophobic cyclohexane portion, in addition to the electron density provided by the tetrazole ring.

In the molecular interaction between **7f** and p53, it could be predicted that the binding mode included a hydrogen-bond-type contact between the ribosidic portion of a guanine (G8) of the co-crystallized DNA fragment in the receptor in addition to two types of interactions, both pi-alkyl, between the tetrazole portion of **7f** and the His179 and Met243 of p53. Two interactions between the bromine atom of the aromatic portion attached to the alpha carbon of the tetrazole were also observed, involving Cys242 and Met 2433; however, due to the types of calculations employed, it cannot be ensured that these interactions are of the halogen interaction type. The interaction between **7f** and p53 could provide insight into the biological activity shown in the in vitro experiments performed. It has been reported that P53/Akt/JNK can regulate proliferation, migration, and invasion in MCF-7 cells [26]. P53 can hamper proliferation, migration, and invasion through the downregulation of Snail, cyclin D1, RhoA, RhoC, and MMP9 at mRNA level and decreased mitogen-activated protein kinase (MEK) and extracellular-signal-regulated kinase (ERK) phosphorylation at the protein level [27]. Therefore, additional tests are needed in order to confirm the p53-mediated inhibition mechanism of cancer cellular processes induced by T-TH compounds.

## 3. Materials and Methods

### 3.1. Chemistry

#### General Information

All reagents, reactants, and solvents were purchased from Merck (Darmstad, Germany) without further purification. Melting points (uncorrected) were determined using a Fisher-Johns melting point apparatus (Cole-Parmer, Vernon Hills, IL, USA). Column chromatography was performed using silica gel (230–400 mesh). Reaction progress was monitored via thin-layer chromatography (TLC) with silica gel plates from Merck (silica gel 60 F_254_), and the spots were visualized under UV light at 254 or 365 nm. Chemical names and drawings were obtained using ChemDraw Professional (version 18.0.0.231). NMR spectra were recorded using a Varian Mercury (400 Mhz) (Varian, Palo Alto, CA, USA) spectrometer, using CDCl_3_ as a solvent and TMS as an internal reference. Chemical shifts were reported as δ values (ppm). Coupling constants J were reported in Hertz (Hz). High-resolution mass spectra (HRMS) were obtained using Brucker microTOF (Bruker Daltonics, Bremen, Germany).

General procedure for compounds **7a–o** (GP) can be found in [8].

### 3.2. Cell Lines

The breast tumor cell lines MCF-7 and HCC1954, CAMA-1, and SKBR-3 were grown in Dulbecco’s Modified Eagle’s Medium (DMEM) (Invitrogen Corporation, Carlsbad, CA, USA) enriched with 6% and 10% fetal bovine serum (FBS). Medium changes and passaging were performed every 3 and 4 days, respectively. The cell lines were kindly provided by Ph.D. Victor Treviño from Tecnológico de Monterrey.

### 3.3. Cell Proliferation Analysis

Cell proliferation was quantified using violet crystal dye in 1× phosphate-buffered saline (PBS) (2.7 mM of KCl, 1.8 mM of KH_2_PO_4_, 136 mM of NaCl, and 10 mM of Na_2_HPO_4_ at pH 7.4). The treated cells were incubated in methanol for 15 min and washed twice with distilled water. Cells were dyed with 0.1% crystal violet and washed three times with distilled water; finally, crystal violet was recovered with 10% acetic acid for analysis in microplate reader Multiskan GO Spectrophotometer (Thermo Scientific™, Ratastic, Finland).

### 3.4. Cell Migration Analysis

Cells were treated with the IC_50_ of T-THs and with mitomycin C (1.2 µg/mL) for 1.5 h. They were subsequently trypsinized, counted (8 × 10^4^ cells/transwell), and seeded into a transwell chamber (upper compartment) before adjusting the final volume to 200 µL of DMEM supplemented with 2% FBS. As a chemoattractant, 600 µL of DMEM supplemented with 10% FBS was used in the lower compartment and incubated for 72 h. The cells were washed with PBS and fixed in 100% methanol. The non-migrating cells were removed from the upper surface of the membrane via scrubbing, followed by staining with 0.1% crystal violet. Afterward, the membranes were incubated with 10% acetic acid and measured using a spectrophotometer (Multiskan GO Spectrophotometer, Thermo Scientific™, Ratastic, Finland) at an optical density of 600 nm.

### 3.5. Cell Invasion Analysis

Cells were treated with the IC_50_ of T-THs and with mitomycin C (1.2 µg/mL) for 1.5 h. They were subsequently trypsinized, counted (8 × 10^4^ cells/transwell), and seeded into a transwell chamber (upper compartment) previously covered with matrigel at 0.1%, before adjusting the final volume to 200 µL of DMEM supplemented with 2% FBS. As a chemoattractant, 600 µL of DMEM supplemented with 10% FBS was used in the lower compartment and incubated for 72 h. The cells were washed with PBS and fixed in 100% methanol. The non-migrating cells were removed from the upper surface of the membrane via scrubbing, followed by staining with 0.1% crystal violet. Afterward, the membranes were incubated with 10% acetic acid and measured using a spectrophotometer at an optical density of 600 nm.

### 3.6. Molecular Docking Studies

Ligand preparation

The chemical structures of 1,5-disubstituted tetrazole-1,2,3-triazoles were modelled as 2D structures using the ChemBio Ultra 12.0 software and converted into 3D structures in MDL format with their protonated states at pH = 7.4 [28]. The geometries of the compounds were calculated using the molecular mechanics force field (MMFF) in Spartan 14 [29]. Finally, using Autodock Tools, the ligands were prepared by inserting polar hydrogens and Gasteiger charges in them as well as rotatable (i.e., single) bonds that were assigned by default, and a PDBQT file was generated [30].

Receptor Preparation

The X-ray coordinates of p53, p58, p38, and JNK1 were retrieved from the Protein Data Bank under the following ID PDB codes: 2AC0, 4BPW, 2ZAZ, and 3ELJ, respectively. Molecular water was removed from the crystallographic structures, and the final preparation and minimization of the receptor structures was carried out by deploying the Dock Prep module of Chimera 1.15 software using the AMBER-ff14SB force field [31]. Lastly, Kollman charges were added to the obtained structure using Autodock Tools, and a PDBQT file was generated.

Docking Calculations

The calculations corresponded to a rigid, blind type of molecule, carried out in Autodock4 using the Lamarckian genetic algorithm [32]. We used grid maps with 120 × 120 × 120 points in the active site of the receptor with the specific coordinates for each receptor (2ACO: x = 28.830, y = −1.308, and z = 10.812; 4BPW: x= 32.816, y = 50.724, and z = −9.296; 2ZAZ: x = 43.860, y = 25.813, and z = 40.917; and 3ELJ: x = 16.423, y = 15.379, and z =23.667). A grid-point spacing of 0.375 Å was applied. AD4.dat parameters were applied to all the ligands. The parameters used were 100 runs, a population size of 150, and a run-termination criterion of a maximum of 27,000,000 generations or a maximum of 250,000 energy evaluations. The visualization and analysis of the nonbonded interactions as hydrogen bonds of the best poses were carried out using Discovery Studio Visualizer v21.1.0.20298 software [33].

## 4. Conclusions

The T-THs presented different grades of inhibition on cell proliferation, migration, and invasion. **7f**, **7l**, and **7g** inhibited the proliferation of the MCF-7 and CAMA-1 and HCC1954 and SKBR-3 cells, respectively. Interestingly, the compounds with a greater effect on migration and invasion inhibition were **7o**, **7b**, **7n**, and **7k** for the CAMA-1, MCF-7, HCC1954, and SKBR-3 cells, respectively. The compounds presented a higher affinity for p53, followed by JNK1, p58, and lastly p38. The T-THs inhibiting proliferation (**7f**, **7l,** and **7g**), migration, and invasion (**7o**, **7b**, **7n**, and **7k**) have an affinity for p53 protein based on the docking studies reported herein, suggesting the T-THs-dependent regulation of p53. The data presented herein support the anticarcinogenic potential of the T-THs that probably target p53.

## Data Availability

All data generated and analyzed during this study are included in this published article and its Appendix A files.

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
