# Peer review of "Proliferation, Migration and Invasion of Breast Cancer Cell Lines Are Inhibited by 1,5-Disubstituted Tetrazol-1,2,3-triazole Hybrids through Interaction with p53"

_molecules, 2023, doi:10.3390/molecules28227600_

Round 1

Reviewer 1 Report

Comments and Suggestions for Authors

The investigation by Moreno-Perea et al., on the anti-proliferative, migration and invasion inhibitory properties of synthesized T-THs against breast cancer  cell lines is not only properly designed but also well-executed. The work presents a refreshing perspective of a group of compounds with growing relevance in the fight against breast cancer. It would be valuable to publish this work, however, a few points are worthy of consideration in order to improve the quality of the work.

-In the Abstract section, the authors should highlight the most promising compound unveilled by the investigation. Also, a sentence should be included at the end stating the significance or potential implication of the findings.

-A robust Conclsions section must be included in the work.

-There are many grammatical errors. A thorough revision/editing should be  performed by a professional.

-Why did the authors not compare their results with a positive control, such as a drug that is already in use for these purposes.

Thank you.

Comments on the Quality of English Language

The manuscript has a number of typograpical and grammatical errors. Authors are encouraged engage the service of a native speaker or professional to thoroughly edit the manuscript.

Author Response

The investigation by Moreno-Perea et al., on the anti-proliferative, migration and invasion inhibitory properties of synthesized T-THs against breast cancer  cell lines is not only properly designed but also well-executed. The work presents a refreshing perspective of a group of compounds with growing relevance in the fight against breast cancer. It would be valuable to publish this work, however, a few points are worthy of consideration in order to improve the quality of the work.

Response: Thank you very much for your valuable comments.

-In the Abstract section, the authors should highlight the most promising compound unveilled by the investigation. Also, a sentence should be included at the end stating the significance or potential implication of the findings.

Response: We added the compounds 7f, 7l and 7g that inhibit proliferation and 7o, 7b, 7n, and 7k for migration and invasion inhibition as well as the importance to study these compounds potential drug for cancer.

-A robust Conclusions section must be included in the work.

Response: The conclusion was robust by highlighting the compounds involved in p53 regulation and its effect in proliferation, migration and invasion.

-There are many grammatical errors. A thorough revision/editing should be  performed by a professional.

Response: The manuscript was doble checked by authors and for native speaker.

-Why did the authors not compare their results with a positive control, such as a drug that is already in use for these purposes.

Response: We compare our results with etoposide and taxol treatment as you suggested.

Reviewer 2 Report

Comments and Suggestions for Authors

In the manuscript submitted, the authors present the anticarcinogenic potential of a series of 1,5-disubstituted tetrazole-1,2,3-triazoles hybrids.

The synthesis of the compounds tested has already been published, and this undermines the significance of the submitted manuscript in a certain way.

The Discussion part of the manuscript needs to be more developed, with a detailed SAR study.

Figure 1 could be reduced to depict a few representative compounds and the remaining part can be moved to SI.  

Author Response

Dear reviewer thank you for your observations and suggestions. 

In the manuscript submitted, the authors present the anticarcinogenic potential of a series of 1,5-disubstituted tetrazole-1,2,3-triazoles hybrids.

The synthesis of the compounds tested has already been published, and this undermines the significance of the submitted manuscript in a certain way.

Response: We agree that the synthesis of compounds is important, however, additional characterization is always needed, therefore, in this paper we contribute with migration, invasion that in many papers are missing. We include 3 additional different cell lines, and it could be noted not all the cell lines respond equally.

The Discussion part of the manuscript needs to be more developed, with a detailed SAR study.

Response: The discussion section was developed with SAR detailed as well as with cell signaling pathways.

Figure 1 could be reduced to depict a few representative compounds and the remaining part can be moved to SI.  

Response: We think the results in Figure 1 are important to show.

Reviewer 3 Report

Comments and Suggestions for Authors

The publication “Proliferation, migration and Invasion of breast cancer cell lines are inhibited by the compounds 1,5-disubstituted tetrazol-1,2,3-triazoles hybrids by interaction with p53” by Marisol Moreno-Perea et al, presents an interesting scientific topic in the field of medicinal chemistry with a wide variety of research conducted on the various cell lines such as CAMA-1, MCF-7, HCC1945 and SKB3-3.

However, there are many issues. I would not recommend this publication to be published in “Molecules”. The publication is poorly written and the results are questionable.

Title: ”Proliferation, migration and Invasion of breast cancer cell lines are inhibited by the compounds 1,5-disubstituted tetrazol-1,2,3-triazoles hybrids by interaction with p53”

Seems quite unclear, I would suggest to modify it.

For instance: ”Proliferation, migration, and invasion of breast cancer cell lines are inhibited by 1,5-disubstituted tetrazole-1,2,3-triazole hybrids through interaction with p53”

Typos/suggestions:

Line 28: “where”, please correct to “were”

Line 35: “both” seems unnecessary.

Line 85: Scheme 1. Should be in bold. Please check throughout the manuscript. Figures, tables and so on. Many mistakes.

Line 99: “synthetized” please correct to “synthesized”

Lines 103-104: “are in needed to be investigated” please revise.

Generally, there are too many formatting issues.

For instance, lines 255-260 and so on. Please check everything carefully.

Figure 2. should be improved. Compound numbers overlap, you should reduce the font size.

Conclusions are very unclear.

All in all, major revision of English language must be done.

In line 99, you state that “Thirteen hybrid compounds were synthetized”, this is very confusing. At first, I’ve noticed that there is no supplementary file.

These compounds are from an older publication: Molecules 2021, 26(20), 6104; https://doi.org/10.3390/molecules26206104

But there is no supplementary file as well. With NMR, HRMS and other spectra. There are no proofs that these compounds are pure or even synthesized. You have no proofs.

Sorry,

Best of luck.

Comments on the Quality of English Language

Poor English. Lots of formatting issues.

Author Response

Dear reviewer thank you for your observations and suggestions.

The publication “Proliferation, migration and Invasion of breast cancer cell lines are inhibited by the compounds 1,5-disubstituted tetrazol-1,2,3-triazoles hybrids by interaction with p53” by Marisol Moreno-Perea et al, presents an interesting scientific topic in the field of medicinal chemistry with a wide variety of research conducted on the various cell lines such as CAMA-1, MCF-7, HCC1945 and SKB3-3.

However, there are many issues. I would not recommend this publication to be published in “Molecules”. The publication is poorly written and the results are questionable.

 Response: I would be grateful  if you argue in detail about questionable results that you are observing.

Title: ”Proliferation, migration and Invasion of breast cancer cell lines are inhibited by the compounds 1,5-disubstituted tetrazol-1,2,3-triazoles hybrids by interaction with p53”

Seems quite unclear, I would suggest to modify it.

For instance: ”Proliferation, migration, and invasion of breast cancer cell lines are inhibited by 1,5-disubstituted tetrazole-1,2,3-triazole hybrids through interaction with p53”

Response: The title was changed as you suggested.

Typos/suggestions:

Line 28: “where”, please correct to “were”

Response: The error was corrected.

Line 35: “both” seems unnecessary.

Response: It was check it.

Line 85: Scheme 1. Should be in bold. Please check throughout the manuscript. Figures, tables and so on. Many mistakes.

Response: The Figures, tables and scheme was changed to bold.

Line 99: “synthetized” please correct to “synthesized”

Response: The error was corrected.

Lines 103-104: “are in needed to be investigated” please revise.

Response: The lines was check it.

Generally, there are too many formatting issues.

For instance, lines 255-260 and so on. Please check everything carefully.

Response: Lines were check it.

Figure 2. should be improved. Compound numbers overlap, you should reduce the font size.

Response: The Figure 2 was corrected as you suggested.

Conclusions are very unclear.

Response: The conclusion was clarify and improved. 

All in all, major revision of English language must be done.

Response: The English language was doble check by authors and for native speaker.

In line 99, you state that “Thirteen hybrid compounds were synthetized”, this is very confusing. At first, I’ve noticed that there is no supplementary file.

Response: The sentence was corrected and a supplementary file was added.

These compounds are from an older publication: Molecules 2021, 26(20), 6104; https://doi.org/10.3390/molecules26206104

But there is no supplementary file as well. With NMR, HRMS and other spectra. There are no proofs that these compounds are pure or even synthesized. You have no proofs.

Response: A supplementary file was added with data of NMR spectra of the compounds used.